# Canadian Snow and Sea Ice: Historical Trends and Projections

Lawrence Mudryk[1], Chris Derksen[1], Stephen Howell[1], Fred Laliberté[1], Chad Thackeray[2], Reinel Sospedra-Alfonso[3], Vincent Vionnet[4], Paul Kushner[5], and Ross Brown[6]

[1]Climate Research Division, Environment and Climate Change Canada, Toronto, Canada
[2]Department of Geography and Environmental Management, University of Waterloo, Canada
[3]Climate Research Division, Environment and Climate Change Canada, Victoria, Canada
[4]Centre National de Recherches Météorologiques, Centre d'Etudes de la Neige, Grenoble, France
[5]Department of Physics, University of Toronto, Toronto, Canada
[6]Climate Research Division, Environment and Climate Change Canada, Montreal, Canada

*Correspondence to*: Lawrence Mudryk (lawrence.mudryk@canada.ca)

**Abstract.** The Canadian Sea Ice and Snow Evolution Network (CanSISE) is a climate research network focused on developing and applying state of the art observational data to advance dynamical prediction, projections, and understanding of seasonal snow cover and sea ice in Canada and the circumpolar Arctic. Here, we present an assessment from the CanSISE network on trends in the historical record of snow cover (fraction, water equivalent) and sea ice (area, concentration, type, and thickness) across Canada. We also assess projected changes in snow cover and sea ice likely to occur by mid-century, as simulated by the Coupled Model Intercomparison Project Phase 5 (CMIP5) suite of earth system models. The historical datasets show that the fraction of Canadian land and marine areas covered by snow and ice is decreasing over time, with seasonal and regional variability in the trends consistent with regional differences in surface temperature trends. In particular, summer sea ice cover has decreased significantly across nearly all Canadian marine regions, and the rate of multi-year ice loss in the Beaufort Sea and Canadian Arctic Archipelago has nearly doubled over the last eight years. The multi-model consensus over the 2020-2050 period shows reductions in fall and spring snow cover fraction and sea ice concentration of 5-10% per decade (or 15-30% in total), with similar reductions in winter sea ice concentration in both Hudson Bay and eastern Canadian waters. Peak pre-melt terrestrial snow water equivalent reductions of up to 10% per decade (30% in total) are projected across southern Canada.

## 1 Introduction

Seasonal terrestrial snow and sea ice influence short term weather and longer term climate by altering the surface energy budget, modifying both the surface reflectivity and thermal conductivity (Serreze et al., 2007; Flanner et al., 2011; Gouttevin et al., 2012). Snow also influences freshwater storage through soil moisture recharge and surface runoff (Barnett et al., 2005). Understanding historical and projected changes to snow and ice is essential both to assess the importance of physical changes to the climate system, and to thus assess their consequent impacts and risks. A previous assessment of the Canadian cryosphere (snow, sea ice, freshwater ice, land ice, frozen ground) was compiled as part of the International Polar Year

(IPY) in 2007-2008, and is described in Derksen et al. (2012). The current study updates the IPY analysis pertaining to terrestrial snow and sea ice by adding nearly a decade of data and including climate model projections of changes over the next 30 to 40 years. This study is focused on Canadian territory, which is nearly completely covered by snow and sea ice for parts of each year with near-continuous coverage over high latitude and high elevation regions. Snow and sea ice are recognized as critical components of Canada's natural environment, ecosystems, and economy. With respect to snow cover, real-time information on the amount of snow on the ground (i.e. depth and water equivalent) is used in operational decision making for water resource planning (Turcotte et al. 2007), snow clearing, evaluation of avalanche risk (Conlan and Jamieson, 2017) and for initialization of Canada's global weather forecast system (Brasnett, 1999). Historical snow cover data are used in a wide range of applications including ecological studies (Luus et al. 2013), water resources (Kang et al. 2014), forest management (Hanewinkel et al. 2008), estimation of snow loads for infrastructure design (Hong and Ye, 2014), impacts on ground frost penetration (Zhang et al. 2008), and for evaluation of climate and hydrological models (Verseghy et al. 2017; Ganji et al. 2017). Snow also makes a significant direct contribution to the Canadian economy through winter recreation (Archambault et al. 2003). With respect to sea ice, knowledge of both historical and future sea ice conditions in Canadian Waters is important for operational ship navigation to ensure economic and safe shipping, particularly in the Northwest Passage. In addition, information on sea ice (e.g. coverage, type and thickness) is required for the initialization and verification of seasonal prediction models (Lindsay et al., 2012; Sigmond et al., 2013). While acknowledging the Canadian focus, much of the applied approach will be extended to other regions of interest through upcoming Coupled Model Intercomparison Project Phase 6 (CMIP6) experiments (Eyring et al., 2016).

Previous studies have shown that warming temperatures, which are amplified at higher latitudes as a natural response to increasing greenhouse gases (Serreze et al., 2009; Pithan and Mauritsen, 2014), reduce the spatial extent and mass of snow and ice (for example, see Derksen et al., 2012). In reality, the linkage between warming temperatures and snow/sea ice reduction is more nuanced due to:

- *regional and seasonal climate variability*: for example, surface temperature warming across Canadian land and ocean areas is not uniform in space and time, but contains regional and seasonal variability driven by natural climatic processes such as the El Niño Southern Oscillation (ENSO) and other oceanic teleconnections, and inter-seasonal and inter-annual changes in the preferred modes of atmospheric circulation (Vincent et al., 2015). The impact of warm temperature departures on the cryosphere vary with season: during spring they are directly linked to the timing and magnitude of melt onset; during fall warm temperatures can be associated with delayed snow cover onset and ice formation, the impacts of which may not become apparent for many months.

- *the selection of cryospheric variables*: the onset, accumulation/growth and melt of snow and sea ice are influenced by many factors. Different metrics are relevant for assessing different impacts on the environment and ecosystems, and these metrics do not always vary coherently with each other (Bokhorst et al., 2016). Changes in snow can be reflected in the timing of snow onset in the fall and melt in the spring, the annual maximum accumulation of snow

mass, or the extent of snow covered area. For sea ice, not only are changes in the fractional ice cover important, but also the type of sea ice present, specifically whether it is thin first-year ice or thicker multi-year ice (Maslanik et al., 2011).

- *other process drivers*: while surface temperature plays a major role in influencing snow and ice, there are other important drivers of change. For instance, increased precipitation in sufficiently cold regions may offset shorter snow seasons (Brown and Mote, 2009). Sea ice dynamics (driven by wind and ocean currents) can play a major role in regional sea ice conditions independent of surface temperature (Howell et al., 2013a).
- *snow/ice forcing of climate anomalies*: variations in snow and ice cover may also generate feedbacks to the atmosphere/ocean circulation that influence climate on seasonal to decadal scales (Cohen et al. 2007, Smith et al. 2010, Scaife et al. 2014), although there is no clear consensus on how changes in Arctic sea ice and snow cover influence midlatitude climate (Francis and Vavrus, 2012; 2015; Francis et al., 2017; Barnes, 2013; Screen et al., 2015).

The first objective of this paper is to provide an overview of observed changes to seasonal terrestrial snow and sea ice across Canadian territory using the longest available time series of validated gridded datasets. We use a multi-data set approach, averaging multiple estimates of terrestrial snow variables together for more robust trends, and using an integrated, multi-source dataset for analysis of sea ice change. The second objective is to compare these recent historical changes to projected changes of snow and sea ice over a similar length of time from the near future out to the middle of the 21$^{st}$ century (2020-2050). We use simulations from state of the art climate models with confidence levels that account for uncertainty in the regional temperature response (note that the focus of this study is not model evaluation – that analysis is described in Kushner et al., 2017). Results are presented in two sub-sections, separated by the observational analysis period (Section 3.1) and the climate projections (Section 3.2). Details on the datasets and methodology are provided in Section 2. We summarize our key findings in Section 4 and present remaining points of discussion in Section 5.

## 2 Data

### 2.1 Historical Datasets

#### 2.1.1 Terrestrial Snow Data

Following Mudryk et al. (2015), we took a multi-dataset approach to analyze observed snow cover change in order to account for observational uncertainty. We calculated monthly snow cover fraction (SCF) and annual maximum snow water equivalent (SWEmax) using daily SWE data taken from the following four sources over the 35 year period from 1981-2015. 1) The Modern-Era Retrospective Analysis for Research and Applications version 2 (MERRA-2), (Reichle et al., 2017, updated from Rienecker et al. 2011), is a National Aeronautics and Space Administration (NASA) atmospheric reanalysis

product generated with the Goddard Earth Observing System Model, version 5.2.0 (GEOS-5) atmospheric general circulation model and atmospheric data assimilation system (ADAS). 2) The temperature index model described by Brown et al. (2003) reconstructs daily SWE using 6-hourly temperature field and 12-hourly precipitation field inputs from ERA-interim reanalysis. This simplified index model includes most of the temperature dependent processes included in the snow component of numerical land surface schemes (e.g., partitioning of precipitation into solid and liquid fractions, melt from rain-on-snow events, specification of new snowfall density, snow aging, and snowmelt). 3) The physical snowpack model Crocus simulates daily SWE using meteorology from ERA-Interim (Brun et al. 2013). 4) The European Space Agency GlobSnow product (version 2; www.globsnow.info) is a gridded product derived through a combination of satellite passive microwave data, forward snow emission model simulations, and climate station observations for non-alpine regions of the northern hemisphere (Takala et al. 2011). The GlobSnow product is masked over regions of complex topography, defined using a high resolution topographic map (5 arcminutes) as regions in which the standard deviation in elevation is larger than 200m. This criterion affects approximately 20% of the NH land surface (principally alpine regions) and a similar percentage of the Canadian land surface. We replaced grid cells containing complex topography with a blend of the GlobSnow data and the mean value from the other three data sources. The weighting for the blend was determined by the fraction of the grid cell area containing complex topography. For grid cells with no complex topography, unaltered GlobSnow data are used. As the fraction of complex topography increases, the weight applied to the GlobSnow data is linearly reduced, reaching zero for grid cells containing only complex topography.

For a given dataset of daily SWE, we interpolated the data to a regular 0.25 degree grid over Canada and applied a 4 mm threshold to produce a daily binary snow cover field. We averaged this daily field over each month to produce a monthly snow cover fraction. Annual maximum SWE fields were calculated as the maximum value of daily SWE attained at each grid location over a given snow season. For both SCF and SWEmax, we computed trends separately for each of the four datasets and then averaged these together into a final trend representing the mean across the observational ensemble.

**2.1.2 Sea Ice Data**

For our analysis of changes in sea ice we examined monthly and seasonal changes in sea ice concentration (SIC) and sea ice thickness. SIC is analogous to our SCF derivation, in that it represents the fraction of the surface that is covered by sea ice but can also be interpreted as the fraction of time over which the surface is fully ice-covered. We extracted total and multi-year ice (MYI) area within Canadian Arctic waters from the Canadian Ice Service Digital Archive (CISDA), which is an integration of a variety of data sets including: satellite observations, surface observations, airborne and ship reports, operational model results and the expertise of ice analysts (see Canadian Ice Service, 2007 and Tivy et al., 2011a for complete details). We selected the CISDA record instead of satellite passive microwave data because 1) the CISDA was found to be more accurate in the shoulder seasons during which passive microwave retrievals can underestimate sea ice concentration by as much as 44% (Agnew and Howell, 2003) and 2) the CISDA sea ice record provides homogeneous data

back to 1968 for regions of the Canadian Arctic (Tivy et al., 2011a), almost 10 years earlier than coverage by satellite passive microwave observations. The CISDA data was analyzed over two historical periods, the 1981-2015 period consistent with available snow data and a longer 1968-2016 period.

Analogous to SWE, which provides a metric for the total amount of snow, we also analyzed maximum landfast sea ice

thickness. We used the Canadian Ice Service record of in situ landfast ice thickness measurements, made regularly at coastal Arctic stations since the early1950's (Howell et al. 2016b).  In general, thickness measurements are available at weekly frequency, starting after freeze-up when the ice is safe to walk on and continuing until breakup or when the ice becomes unsafe. Details of this dataset are provided by Brown and Cote (1992) and are available on the Canadian Ice Service Web site at http://iceglaces.ec.gc.ca.

**2.1.3 Surface Temperature**

In light of documented differences in gridded temperature datasets over Canada (Rapaic et al., 2015), surface temperature trends were derived from a blend of six reanalysis products: the European Centre for Mid-Range Weather Forecasting (ECMWF) Reanalysis (ERA-Interim, Dee et al., 2011), the Japanese 55-year and 25-year Reanalyses (JRA-55, Kobayashi et al., 2015; JRA-25, Onogi et al, 2007), the Modern-Era Retrospective analysis for Research and Applications version 1 and 2

(MERRA-1, Reinecker et al., 2011; MERRA-2, Reinecker et al., 2017), and the Climate Forecast System Reanalysis (CFSR, Saha et al., 2010). The average trend was determined by first computing the average of JRA-55 and JRA-25 (resulting in a single JRA trend), and the average of MERRA-1 and MERRA-2 (resulting in a single MERRA trend). The multi-reanalysis mean trend was computed by averaging (ERA-Interim, CFSR, JRA, and MERRA). Reanalysis was used instead of point station data in order to produce spatially continuous trends over both terrestrial and marine areas.

**2.2 Model Simulations**

**2.2.1 Terrestrial Snow**

We used monthly mean output from the suite of historical and future simulations from the CMIP5 archive (Taylor et al. 2012; http://cmip-pcmdi.llnl.gov/cmip5/) to evaluate SCF (denoted as 'snc' in CMIP5 output) and SWE (denoted as 'snw' in CMIP5 output). The models utilized for snow analysis are listed in **Table 1;** note that no model selection was performed –

all models with archived snow data were utilized. The resolution of available atmospheric/land model output ranges from approximately 1.3°x0.9° to 2.8°x2.8° longitude and latitude. Snow projections were selected from the Representative Concentration Pathway (RCP) 8.5 projected forcing scenario because it most closely resembles the observed emissions pathway over the past decade (Peters et al. 2013). We then computed individual trends for each realization and then took the inter-realization average across each model to calculate individual model ensemble means. These values were averaged to

determine the CMIP5 multi-model mean values. If only a single realization was available, that was used directly as input to

the multi-model mean calculation. SWE and SCF output were also taken from a large initial condition ensemble (50 realizations) of the Second Generation Canadian Earth System Model (CanESM2; Arora et al., 2011), a global earth system model from the Canadian Centre for Climate Modelling and Analysis. Each of the realizations of this ensemble evolves over the 1950-2100 period under identical historical radiative forcings in accordance with CMIP5 from 1950-2005 and the

165 RCP8.5 scenario from 2006-2100 (see Thackeray et al, 2016 for more details). As such, differences among realizations result only from differences in the initial climate state and are due to natural variability alone). This ensemble was used to characterize the role of internal climate variability on projected snow cover changes over Canada.

### 2.2.2 Sea Ice

Monthly mean sea ice concentration (denoted as 'sic' in CMIP5 output) and the land-sea mask (denoted as 'sftlof' in CMIP5

output) were also retrieved as available from CMIP5 output resulting in a sea ice ensemble that comprises 42 models and a combined total of 91 simulations (Table 2). The resolution of ocean/ice model grids are generally equal to or finer than corresponding atmospheric grids ranging from approximately 0.4°x0.4° to 1.0°x1.0° longitude and latitude. The sea ice concentration was projected to the EASE grid using the same procedure as in Laliberté et al. (2016). For projections of sea ice-free conditions over the four Canadian marine subregions (Baffin Bay, Beaufort Sea, Canadian Arctic Archipelago, and

Hudson Bay), we excluded models that do not capture at least 75% of the region's observed ocean area (resulting in about half of the models rejected in the Canadian Arctic Archipelago) and compute sea ice extent as in Laliberté et al. (2016).

### 2.3 Evaluation of Trend Significance

In order to make comparable significance calculations for the observed and projected trends, we use a Monte Carlo method following Swart et al. (2015) and applied in Howell et al. (2016b). The method ensures that spread due to internal variability

is comparable for all CMIP5 models, even those that only include a single realization in their archived output. This is achieved by adding simulated noise representing internal variability to those models with only a single realization. The added noise is calculated from the collection of models with multiple realizations under the assumption that the spread due to internal variability is the same across all models. A trend is significantly different from zero if it is distinguishable from the combined spread due to interannual variability (using a t-distribution for each simulation), internal variability (with noise

added as described), and model spread.

We use an analogous approach to compute significance for the various observation-based dataset trends. We assume that differences in the trends arise from differences in retrieval performance and reanalysis methodologies but not from sampling of internal variability. Thus, a trend is significantly different from zero if it is distinguishable from the combined spread due

to interannual variability (using a t-distribution for the dataset mean) and the added spread due to differences in the trends among the different data sets.

## 3 Results

### 3.1 Observed Trends in Terrestrial Snow and Sea Ice

Seasonally averaged trends in terrestrial snow cover fraction (SCF) and sea ice concentration (SIC) over the 1981-2015 time period are shown in Fig. 1 (the dashed line denotes limit of Canadian marine territory). Throughout the paper, we present seasonal trends for permutated months in order to more closely match the mid-season peaks of SCF (January), SWE (February-March) and SIC and sea ice thickness (March). Positive trends in SCF (more snow cover) are evident over a small region of the southern prairies in winter (January/February/March, JFM) and more extensively over western Canada in spring (April/May/June, AMJ). Trends in all remaining regions and seasons are negative, notably over eastern Canada in spring and most of the Canadian land area in the fall (October/November/December, OND). The predominantly negative trends in snow cover are consistent with previous studies (Brown and Braaten, 1998; Vincent et al. 2015) but with evidence of a shift to stronger snow cover reductions in the snow onset period over eastern Canada in response to enhanced OND warming shown in Fig. 2 **(discussed below)**. The observed rates of snow cover change are also consistent with the recent Snow, Water, Ice, Permafrost in the Arctic (SWIPA) assessment (Brown et al. 2017) that documented annual SCD changes over Arctic land areas of -2 to 4 days/decade (~ -1 to -2% per decade assuming 250 days mean snow cover). Other studies focused on Arctic snow cover (i.e. Derksen and Brown, 2012; Derksen et al., 2016) have identified spring snow cover losses that are stronger than those in Fig. 1. This difference may stem from stronger spring trends in the NOAA snow chart data record (the NOAA dataset was not used in this study due to known deficiencies in the fall period; Brown and Derksen, 2013) compared to other snow products (Mudryk et al., 2017), and stronger trends across the Eurasian Arctic compared to North America (Derksen et al., 2016).

SIC trends over Canadian waters for this period are almost exclusively negative in all seasons. Regions with the strongest SIC declines are eastern Canadian waters in winter and spring, and the Canadian Arctic Archipelago and Hudson Bay in summer and fall consistent with the warming patterns shown in Fig. 2. The SCF and SIC trends can be viewed collectively as changes in the timing and extent to which highly reflective snow and ice cover the Earth's surface, with important implications for the surface energy budget. There are no sharp boundaries in trends across adjacent land and ocean regions. This provides confidence in the consistency of the snow and ice datasets, as well as evidence of a coherent response of snow and ice to temperature forcing across terrestrial and marine regions.

Figure 2 shows seasonal surface air temperature (TAS) trends over the 1981-2015 period computed from a blend of six atmospheric reanalysis datasets (see Section 2.1.3). TAS trends are generally positive, although no trends are seen throughout the northwestern portion of the country during winter and spring, and there is significant cooling over the Canadian prairies in spring. Land areas with cooling trends are co-located with positive SCF trends in Fig. 1, although the region with positive SCF trends is slightly more extensive. The winter and spring season cooling over northwestern and

central Canada is consistent with the influence of North Pacific oceanic variability over the last 35 years (Mudryk et al., 2014). Climate in this region (including temperature and precipitation, and hence snow cover) is strongly influenced by the Pacific Decadal Oscillation (a pattern of sea surface temperatures and associated sea level pressure changes in the North Pacific); observed sea surface temperature trends have been negative over the last 35 years consistent with reduced warming and increased snow evident in the SCF trend patterns presented here. The reanalysis TAS trends are also seasonally and spatially consistent with the analysis of Rapaic et al. (2015, Fig. 13) based on blended data from both homogenized station observations and multiple reanalyses. In both reanalysis and in situ data, Arctic trends are strongest in the fall and winter, with an increase in the magnitude of warming from the southwest to northeast regions of the country apparent in all seasons.

To examine the role of air temperature trends in driving the observed snow and ice cover trends, spatial correlations (centered and uncentered) were calculated to quantify the pattern relationship between SCF, SIC, and TAS trends (Fig. 3). The centered (uncentered) statistic measures the similarity of the two patterns after (without) removal of the domain mean. A large negative uncentered correlation indicates that the correlation between the two fields is negative on average, but does not require that the field patterns are congruent. A large negative centered correlation does require spatial similarity in the field patterns. Both SCF and SIC trend show large uncentered correlations for all seasons indicative of the general relationship between increasing temperatures and decreasing SCF and SIC. During JFM and AMJ, the large centered correlations between SCF and TAS indicate that the spatial patterns are also similar, and hence there is a strong association between SCF and TAS trends at the local scale during these seasons, with reduced connections during JAS and OND (consistent with Mudryk et al, 2017). SIC trend patterns are more closely associated with warming patterns during ice onset/growth (OND and JFM), however overall there is less co-variability of SIC trends with TAS trends than for SCF trends. This difference may stem from the fact that ice (especially MYI) melts more slowly than snow, and the additional influence of dynamical effects (such as wind-driven redistribution of sea ice) weakens the thermodynamically driven spatial association between SIC and TAS trends.

Trends in SWEmax, which indicates water resource and streamflow potential just before spring melt, are shown in Fig. 4. The figure represents averages from the same datasets used to generate the SCF trends in Fig. 1. Trends are negative over much of Canada, indicating a reduction over time in SWEmax at the onset of the melt season each spring. Because SWE varies with the amount of accumulated snowfall, we may expect a weaker relationship to surface temperature and a stronger connection to precipitation trends. Figure 5 shows trends in annual snowfall estimated from the CANGRD dataset (Milewska et al., 2005) based on interpolated adjusted station data from Mekis and Vincent (2011) and monthly rain/snow fraction obtained from ERA-interim 6-hourly 2m air temperature data assuming a 0°C threshold for rain/snow separation. The changes in snowfall are broadly consistent with the changes in SWE over much of Canada. Notable exceptions include the band of increased snowfall over the Northwest Territories and western Nunavut (which shows negligible or decreasing SWE trends) and the southern portion of Ontario/Québec which shows strongly decreasing SWE trends; however, both of these

regions have experienced stronger warming trends over the full snow season than the western provinces. While the changes
in snowfall are generally consistent with the observed changes in SWE, it is unclear to what extent regional snowfall changes
are themselves correlated with local temperature changes, either due to increased melt during winter thaw events or long
term trends in the solid fraction of precipitation (for the latter see Vincent et al., 2015). We return to this discussion point at
the end of Section 3 and again in the discussion of Section 4.

The longer period of consistent summer sea ice information in Canadian waters from the CISDA allows consideration of
additional years not covered by the 1981-2015 trends in Fig. 1. To be consistent with Tivy et al. (2011a) and Derksen et al.,
(2012) the summer ice season is defined as average sea ice area from June 25 to October 15 for the Beaufort Sea, CAA, and
Baffin Bay regions, and from June 18 to November 19 for Hudson Bay, Hudson Strait, Davis Strait and Labrador Sea.
Between 1968 and 2016, sea ice area averaged over the summer period has experienced significant decreases in almost every
region of the Canadian Arctic, up to 20% per decade in some regions (e.g. the Hudson Strait and Labrador Sea; Fig. 6).
Compared to previous trends reported by Tivy et al. (2011a) over the period of 1968-2008 and Derksen et al., (2012) for
1968-2010, more regions are now experiencing significant decreases and the rate of decline is stronger in all regions except
Hudson Bay (Fig. 7e). For MYI, there are more regions across the Canadian Arctic, particularly in the western CAA, that are
now experiencing significant declines compared to previous studies (e.g. Tivy et al., 2011a; Derksen et al., 2012; Fig. 6).
The largest declines in MYI occurred in the CAA and Beaufort Sea, both of which almost doubled their rate of decline for
1968-2016 when compared to the trend for 1968-2008 (Fig. 7e).

A stepwise reduction in Hudson Bay sea ice area occurred in the mid-1990s (Fig. 7d; Tivy et al., 2011b; Hochheim and
Barber, 2014) and Baffin Bay has experienced consistently low sea ice area since 1999 (Fig. 7c); whereas, considerably
more interannual variability is apparent in the Beaufort Sea and CAA (Figs. 7a and 7b). Of note, the Beaufort Sea
experienced a record low sea ice area in 2012, becoming virtually ice-free near the end of the melt season (Babb et al.,
2016). This was nearly repeated in 2016. As previously reported, the CAA eclipsed the previous and long standing record
low ice year of 1998 in both 2011 and 2012 (Howell et al., 2013b). A contributing factor to the decline of sea ice across the
Canadian Arctic is increasing spring air temperature (see Fig. 2) coupled with longer melt seasons resulting in the absorption
of more solar radiation and increased ice melt (Howell et al., 2009; Tivy et al., 2011a; Stroeve et al., 2014; Parkinson, 2014).

Arctic sea ice thickness has declined in recent years (e.g. Kwok and Rothrock, 2009; Haas et al., 2010; Laxon et al., 2013;
Richter-Menge and Farrell, 2013; Kwok and Cunningham, 2015; Tilling et al., 2015). These studies indicate thickness
declines are greater in the Beaufort Sea compared to the north facing coast of the CAA, which still contains some of the
thickest sea ice in the world. Unfortunately, the spaceborne sensors used to obtain sea ice thickness information are not of
sufficient spatial resolution to provide reasonable thickness estimates within the CAA. However, the Canadian Ice Service
record of in situ landfast ice thickness measurements represents one of the longest datasets in the Arctic, and spans over 5

decades (Howell et al., 2016b). However, the seasonal behaviour of landfast ice thickness can provide useful information for understanding the interannual variability because ice growth is almost entirely due to thermodynamic forcing. Significant declines in maximum ice thickness have occurred at 3 sites in the CAA (Cambridge Bay, Eureka, and Alert) with decreases ranging between -3.6 to -5.1 cm per decade over the period from the late 1950s to 2016 (Fig. 8). No significant trend was found at Resolute but an early study from Brown and Cote (1992) reported a significant increase in maximum ice thickness at Resolute over the period from 1950-1989.

Although systematic measurements at other regions within the CAA are unavailable or contain too much uncertainty, surveys in 2011 and 2014 of ice thickness from airborne electromagnetic induction (AEM) described by Haas and Howell (2015) indicated the ice is still reasonably thick: the mode of the measured thickness distribution was 1.8-2.0 m, and sea ice between 3-4 m in mean thickness was found in the MYI regions throughout the CAA.

### 3.2 Projected Changes in Terrestrial Snow and Sea Ice

Projected trends in SCF and SIC for the 2020-2050 time period across Canadian territory are shown in Fig. 9. These projections are the multi-model mean from the ensemble of CMIP-5 climate models, using the RCP8.5 forcing scenario (which assumes 'business as usual' continued growth of greenhouse gas emissions through the 21st century). While other scenarios exist, projections to the mid-century are primarily dependent on natural variability and model dependent uncertainties rather than the choice of forcing scenario. For example, Figs. 4 from Hawkins and Sutton (2009, 2011) suggest that scenario uncertainty contributes less than 10% of the total uncertainty (natural variability, model uncertainty and scenario uncertainty) for regional-scale, decadal mean temperature at a lead time of thirty years, and substantially less than 10% for regional-scale precipitation. The multi-model projected mean changes in surface temperature are positive in all seasons, hence only reductions in ensemble mean SCF and SIC are evident in Fig. 9. Seasonal differences and varying sensitivity of snow and ice to temperature forcing drives the spatial variability seen in Fig. 9. During winter, projected snow cover reductions are greatest along the southern margins of Canada, where temperature increases directly result in less snow. Temperatures remain sufficiently cold at higher latitudes (despite projected warming) so there is no projected response in this region in JFM SCF. During spring, the region of snow sensitivity to temperature forcing shifts north, across the boreal forest, subarctic, and Arctic tundra, which leads to the negative SCF trends projected across these regions during AMJ. Ensemble-mean reductions projected for SIC are very strong and focused on the ice melt (summer) and ice formation (fall) seasons, with the exception of Hudson Bay and eastern Canadian waters which also have projected winter season loss of sea ice cover.

While natural decadal-scale climate variability resulted in cooling trends (and hence positive SCF and SWEmax trends) for some regions and seasons during the 1981-2015 period (see Figs. 1, 2 and 4), over the longer 1948-2012 period observed surface temperature trends over Canada are almost exclusively positive (Vincent et al., 2015). Similarly, we expect there

could be short term, localized fluctuations in trend direction and magnitude from the projections shown in Fig. 9. These localized fluctuations exist in individual realizations of the model projections (not shown) and act to increase the spread in the TAS, SCF, and SIC responses. This spread reduces the significance of the ensemble-mean response for a given confidence level; for this reason only the coastal regions in British Columbia show significant SCF decreases at the 90% confidence level, whereas projections show widespread reductions in SIC at the 90% confidence level. The differences between SCF and SIC in the significance of their projected responses may result from varying sensitivity to temperature forcing if SCF depends more strongly on the local temperature response than SIC.

Figure 10 shows trends in SWEmax also derived from the CMIP-5 multi-model ensemble. The ensemble mean shows that SWEmax loss will be extensive (5-10% per decade through 2050, or a cumulative 15-30% reduction over the entire 2020-2050 period) over much of Alberta and British Columbia, and similarly in Southern Ontario and the Maritime provinces (note that the greatest near-future loss in the CMIP5 ensemble occurs just south of the Canadian border, not shown in Fig. 10). While positive Arctic SWEmax trends start to emerge by mid-century in the Eurasian Arctic (not shown; see Brown et al., 2017) minimal change is projected across high latitude land areas of Canada. This may result because increasing temperature (which shortens the snow accumulation season) balances projected increases in snowfall.

Month by month projected changes in Canadian snow cover extent (total area of snow cover summed over the Canadian land region) and snow mass (determined by multiplying the density of water by the total volume of SWE summed over the Canadian land region) for the CMIP5 multi-model ensemble, and the large initial condition ensemble from the second generation Canadian Earth System Model (CanESM2) are shown in Fig. 11. The two ensembles agree that the greatest near-future snow loss (as a percentage of climatological snow) is projected to occur in the shoulder seasons (Oct-Nov, May-Jun). During mid-winter there is minimal percentage change in snow cover extent projections because winter temperatures over northern regions of Canada will remain cold enough to sustain snow cover and there is greater climatological snow extent in winter, which results in smaller percentage changes reflected in Fig. 11. The projected trends from CMIP5 are similar in magnitude to the rate of change during the historical period considered in this study, while trends from CanESM2 are slightly stronger due to greater warming in CanESM2 compared to the CMIP5 multi-model mean (Thackeray et al., 2016). Because the CMIP5 ensemble includes many different models, there is a greater spread in most months than the CanESM2-LE, which represents multiple climate realizations generated from a single model. Nonetheless, it's striking that the interquartile range of the CanESM Large Ensemble is quite comparable to that of the CMIP model ensemble. This likely indicates that a large portion of the CMIP5 inter-model spread is associated with internal variability, a similar result to Mudryk et al., (2017).

Sospedra-Alfonso and Merryfield (2017) showed that the interannual variability in monthly SWE between January and April can be skillfully estimated with a multiple linear regression model based on precipitation and temperature predictors. This

statistical model suggests coherent relationships that provide context to the character of the changes to terrestrial snow cover and SWE that are being considered here. An important aspect of their analysis is that mutual correlations between precipitation and temperature variability are accounted for when analyzing their effects on SWE variability. One of their key results was that regimes where SWE variability is dominated by temperature variability and those where it is dominated by precipitation variability can be estimated using a diagnostic temperature metric. This diagnostic temperature ($T_d$) is illustrated for the month of March in Fig. 12 for both present-day (1981-2015) and mid-century projections (2016-2050). The diagnostic temperature is approximated by the average near surface air temperature averaged over the consecutive months from snow onset up to the month under consideration. A temperature threshold, $T_{th}$, divides March SWE into two regions characterized by their sensitivity to precipitation and temperature variability; the region where $T_d < T_{th}$ has March snowpack variability driven mainly by precipitation, whereas the region where $T_d > T_{th}$ has temperature-driven March snowpack variability. The division between these two regions differs depending on the month under consideration (for example, the extent of the precipitation dominated region decreases during spring when the diagnostic temperature isotherms shift northward), and the temperature threshold varies slightly depending on the time period under consideration but is generally found between $T_{th} = -5 \pm 1°C$. See Sospedra Alfonso and Merryfield (2017) for further details.

In 1981-2015, regions with temperature-dominated March snowpack ($T_d \gtrsim -5°C$) include coastal British Columbia, southern Alberta, southwest Saskatchewan, southern margins of the St. Lawrence River and the Great Lakes, the Maritime provinces and Newfoundland. Because of the projected warming, the seasonal isotherms shift northward and toward higher elevations extending the portion of the snow cover that is more sensitive to temperature variations. By 2016–2050, regions with temperature-driven snowpack also include most of southern Canada, the interior of British Columbia, and more extensive portions of the Prairies provinces, Ontario and Québec. In these regions, March SWE is expected to decrease during anomalously warm years. The portion of the snow cover with $T_d < -20°C$, which is largely unaffected by temperature variability (Sospedra-Alfonso and Merryfield, 2017) and encompasses areas of the Northwest Territories and most of Nunavut in present-day climate, is projected to recede to the Canadian Archipelago by mid-century.

Turning to projected changes in sea ice, Fig. 13 shows the observed record of September sea ice extent (sea ice concentration > 15%) across Canadian marine regions, compared to the CMIP-5 multi-model ensemble. While the observed and simulated trends are similar, only those years with strong observed negative departures from the long term trend (e.g. 1998 and 2012) reach the mean simulated ice extent. During the historical record, the models have a large spread (gray histogram, right side of Fig. 13), with most models within the range of observations. After 2050, the large spread persists, but most models are ~10% below the historical sea ice extent (yellow histogram, right side of Fig. 13).

The probability of sea ice free conditions by 2050 for regions of the Canadian Arctic calculated from the CMIP-5 multi-model ensemble are shown in Fig. 14. SIC from each model realization is rescaled to account for differences in relative land

and ocean fraction in a given region. For a given month, model and ice area threshold, a particular region is considered sea ice free if more than 94% of its component grid cells have SIC below the ice area threshold for five out of the six years following 2050. The value of 94% is equivalent to the criterion established in Kirtman et al. (2013) to denote sea ice free conditions for the Arctic as a whole (less than $1\times10^6$ km$^2$ of sea ice in the Arctic ocean). Probabilities are calculated as the fraction of total realizations with ice-free conditions for a given region (see Laliberté et al., 2016, for a full description of the sea ice free probability methodology). Use of two ice area thresholds, 5% and 30%, applied to each grid cell, indicates the sensitivity in timing to the definition of minimum ice area. Under a 5% ice area scenario, there is a greater than 50% probability that all Canadian regions will be sea ice free in September by the year 2050. Ice free probabilities are similar for August, but lower for October and November. Hudson Bay, which is already largely ice free in August and September, has a high probability of being ice free for 4 consecutive months. With a more relaxed threshold of 30% ice area, probabilities are (by definition) greater for all regions and months. By mid-century, Baffin Bay is projected to be ice free for August through October, and 2 months of ice free conditions in the Beaufort Sea and the CAA are a possibility.

## 4 Key Findings

This assessment of observed historical changes in terrestrial snow cover and sea ice over Canada, together with projected changes to the middle of the 21st century, has produced the following key findings:

*Historical datasets show the fraction of Canadian land and marine areas covered by snow and ice is decreasing.*
Observations show decreased SIC in all seasons and decreased terrestrial SCF in fall (delayed snow cover onset) and spring (earlier snow melt). There is regional and seasonal variability in the direction and strength of the trends (for example, some increases in spring snow cover across boreal western Canada) due to seasonal and spatial variability in surface temperature trends resulting from natural climate variability. There is evidence of decreasing annual maximum SWE (reflective of shallower snow depth) consistent with the study of Mudryk et al. (2015) and trends of annual maximum snow depth reported in Vincent et al. (2015). There is only regional evidence (western CAA) of increasing winter season snow accumulation and hence higher SWE across Arctic Canada. Summer season total ice cover is decreasing significantly across nearly all Canadian marine regions. MYI losses are greatest in the Beaufort Sea and the western CAA. In just 8 years, the rate of MYI loss nearly doubled over the 1968-2016 period, compared to a previous assessment over 1968-2008. 60 year long records of in situ landfast ice thickness show evidence of thinning ice in the CAA that was not evident in an earlier study by Brown and Cote (1992), which covered the late 1950s to 1989.

*Canadians should anticipate further reductions in snow and sea ice cover by the middle of the 21st century.*

Averaging projections across many climate model simulations provides evidence that SCF, SWEmax, and SIC will continue to decrease across Canadian land and marine areas through the middle of the 21st century. However, this decrease need not be spatially uniform and regions of negligible decrease or even increase are possible in the near future due to climate variability competing with anthropogenic forcing at the decadal and multi-decadal time scale. For the highly populated regions of southern Canada, there is evidence of a shift in the primary control on inter-annual variability in snow cover from

a regime dominated by precipitation to one dominated by surface temperature. While the thickest sea ice in the world will continue to be present in the CAA and along its northern coast, climate models suggest that Canadian Arctic marine regions which are currently ice covered could be sea ice-free in the summer by 2050.

## 5 Discussion

Snow cover is a defining characteristic of the Canadian landscape for a few months each winter along the southern margins

of the country and up to 9 or 10 months each year in the high Arctic, evolving from nearly complete snow cover over the entire country in the winter to a near total loss of snow cover by the summer. Highly reflective snow cover acts to cool the climate system, effectively insulates the underlying soil, and stores and redistributes water in solid form through the accumulation season before spring melt. Sea ice insulates the ocean from the atmosphere; provides an essential habitat for northern mammals, influences navigation and access to the north, and is of high importance to the traditional lifestyle of

northern communities. This assessment of observed and projected changes in seasonal terrestrial snow and sea ice is focused on Canadian territory, but a number of cross-cutting issues with broader implications for understanding interactions between the cryosphere and climate system were identified through this analysis:

1. The majority of previous assessments of snow cover trends (i.e. Derksen and Brown, 2012; Hernandez-Henriquez et al,

2015; Derksen et al., 2016; Kunkel et al., 2016) were based on the NOAA snow chart climate data record (NOAA-CDR; Estilow et al., 2015) selected on the basis of the longest available record. The reliance on individual datasets such as the NOAA-CDR, however, makes trends prone to uncertainty due to the inherent uncertainties in an individual dataset (for instance, see Brown and Derksen, 2013 and Mudryk et al., 2017 for issues related to the NOAA-CDR). It is clear that consideration of multiple datasets either as a means of showing the range of trends from individual datasets (Mudryk et al.,

2017), for calculating confidence intervals around an individual dataset (Brown and Robinson, 2011), or to benchmark other datasets (Brown et al., 2010; Hori et al., 2017) is a more robust approach. However, in this case, we acknowledge that the availability and use of multiple datasets came at the sacrifice of time series length (the NOAA record extends back to 1967, whereas passive microwave satellite data and reanalyses such as ERA-interim only begin in 1979).

2. Alpine snow poses a unique challenge to trend analysis because the coarse spatial resolution of gridded products used for climate analysis cannot resolve the high degree of spatial variability (driven by land cover variability and steep topographic

gradients) in alpine regions. Wrzesien et al. (2018) have shown that blended gridded products like the one used in this analysis may significantly underestimate SWE in alpine regions. Because alpine show trends vary with elevation in a complex manner (Sospedra-Alfonso et al., 2015; Hamlet et al., 2005), trends from coarse resolution products like the one

used in our analysis are comparatively more uncertain in alpine areas compared to other regions. It is imperative that we address and improve our ability to characterize variability and change in alpine snow because it is these regions that are both extremely sensitive to climate induced changes in snow accumulation (i.e. elevation dependent changes in rainfall versus snowfall ratios) and impactful with respect to water resources (Fyfe et al., 2017; Berg and Hall, 2017; Sospedra-Alfonso et al., 2015; Scalzitti et al., 2016).

3. Changes in sea ice are driven by warming temperatures, but also by changes in atmospheric circulation. The Beaufort Sea was once a region where ice would thicken and age before being transported to the Chukchi Sea and re-circulated in the Arctic (Tucker et al., 2001; Rigor et al., 2002) but now the region has become a considerable contributor to the Arctic's MYI loss (Kwok and Cunningham, 2010; Maslanik et al., 2011; Krishfield et al., 2014; Galley et al., 2016). Ice is still being

sequestered from the Canadian Basin and transported through the Beaufort Sea during the summer months but the ice is now younger and thinner and unable survive the melt season en route to the Chukchi Sea (Howell et al., 2016a). The CAA was also a region with historically heavy MYI conditions present throughout the melt season but ice conditions have become lighter in recent years (see Fig. 7b). The replenishment of CAA MYI via first year ice aging and MYI inflow from the Arctic Ocean has decreased in recent years because of increased temperature and changes in atmospheric circulation (Howell et al.,

2015).

4. There is a strong association between the magnitude of warming and snow and ice loss both in observational datasets and climate model simulations, with projected declines in snow and sea ice cover proportional to the amount of future warming (Thackeray et al., 2016; Mudryk et al., 2017; Notz and Stroeve, 2016). The multi-model mean projections indicate

decreasing snow and ice cover because the multi-model mean projects a warmer climate by mid-century. Within this multi-model mean, however, individual climate model realizations contain regions and seasons with cooling trends (see Mudryk et al., 2014). It is important to remember, therefore, that we live in a single climate realization, while models produce dozens of potential realizations of a future climate. The multi-model mean warming trend (with associated reductions in snow and ice cover) is indicative of a high likelihood of a warmer future, but there will be decadal-scale natural variability, particularly at

regional scales, projected onto this overall trend.

The objective of this paper was to provide a physical climate assessment of observed and projected changes in snow and ice across Canada. While not the focus of this study, these changes will have profound impacts on terrestrial and marine ecosystems, and many sectors of the Canadian economy. This includes risks related to freshwater supply from snow (Sturm

et al 2017) and other impacts of changing snow on the Canadian landscape and economy (Sturm et al., 2016). Accurately

estimating dates of summer sea ice-free conditions in Canadian regions (Laliberté et al. (2016) has important implication for climate studies, but also for determining impact and mitigation strategies. For example, the decreases in MYI within the Beaufort Sea and the CAA illustrated in Fig. 6 were found to be statistically linked to an increase in shipping activity, pointing out the potential implications of continued sea ice declines in these regions (Pizzolato et al., 2016).

**Data Availability**

Environment and Climate Change Canada's Canadian Centre for Climate Modelling and Analysis executed and made publically available the CanESM2 Large Ensemble simulations used in this study. Eric Brun provided data from the Crocus
snowpack model, and Ross Brown provided the Brown data set. These data are available from the paper's authors upon request. Other observation-based data sets are available for download from their respective data centers or upon request from the authors.

**Acknowledgements**

This work is a core contribution of the Canadian Sea Ice and Snow Evolution Network (CanSISE), a climate research network focused on developing and applying state of the art observational data to advance dynamical prediction, projections, and understanding of seasonal snow cover and sea ice. We acknowledge funding from the Natural Sciences and Engineering Research Council of Canada's Climate Change and Atmospheric Research Initiative via the CanSISE Network. The original
manuscript was improved thanks to comments from C. Marty and one anonymous reviewer.

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

**Table 1: List of CMIP5 models (with number of realizations) used for terrestrial snow analysis of both historical and future periods.**

| Model | No. |
|---|---|
| BCC-CSM1.1 | 1 |
| BNU-ESM | 1 |
| CanESM2 | 5 |
| CCSM4 | 6 |
| CNRM-CM5 | 5 |
| CSIRO-Mk3.6.0 | 10 |
| FGOALS-g2 | 1 |
| GISS-E2-R | 1 |
| INM-CM4 | 1 |
| MIROC5 | 3 |
| MIROC-ESM | 1 |
| MPI-ESM-LR | 3 |
| MRI-CGCM3 | 1 |
| NorESM1-ME | 1 |
| NorESM1-M | 1 |

**Table 2: List of CMIP5 models (with number of realizations) used for sea ice analysis of both historical and future periods.**

| Model | No. | Model | No. |
|---|---|---|---|
| BCC-CSM1-1 | 1 | MIROC5 | 3 |
| BCC-CSM-1-m | 1 | HadGEM2-CC | 1 |
| BNU-ESM | 1 | HadGEM2-ES | 4 |
| CanESM2 | 5 | MPI-ESM-LR | 3 |
| CMCC-CESM | 1 | MPI-ESM-MR | 1 |
| CMCC-CM | 1 | MRI-CGCM3 | 1 |
| CMCC-CMS | 1 | MRI-ESM1 | 1 |
| CNRM-CM5 | 5 | GISS-E2-H | 1 |
| ACCESS1.0 | 1 | GISS-E2-H-CC | 1 |
| ACCESS1.3 | 1 | GISS-E2-R | 5 |
| CSIRO-Mk3.6.0 | 10 | GISS-E2-R-CC | 1 |
| FIO-ESM | 1 | CCSM4 | 6 |
| EC-EARTH | 11 | NorESM1-M | 1 |
| inmcm4 | 1 | NorESM1-ME | 1 |
| IPSL-CM5A-LR | 4 | HadGEM2-AO | 1 |
| IPSL-CM5A-MR | 1 | GFDL-CM3 | 1 |
| IPSL-CM5B-LR | 1 | GFDL-ESM2G | 1 |
| FGOALS-g2 | 1 | GFDL-ESM2M | 1 |
| FGOALS-s2 | 1 | CESM1(BGC) | 1 |
| MIROC-ESM | 1 | CESM1(CAM5) | 3 |
| MIROC-ESM-CHEM | 1 | CESM1-CAM5-1-FV2 | 1 |

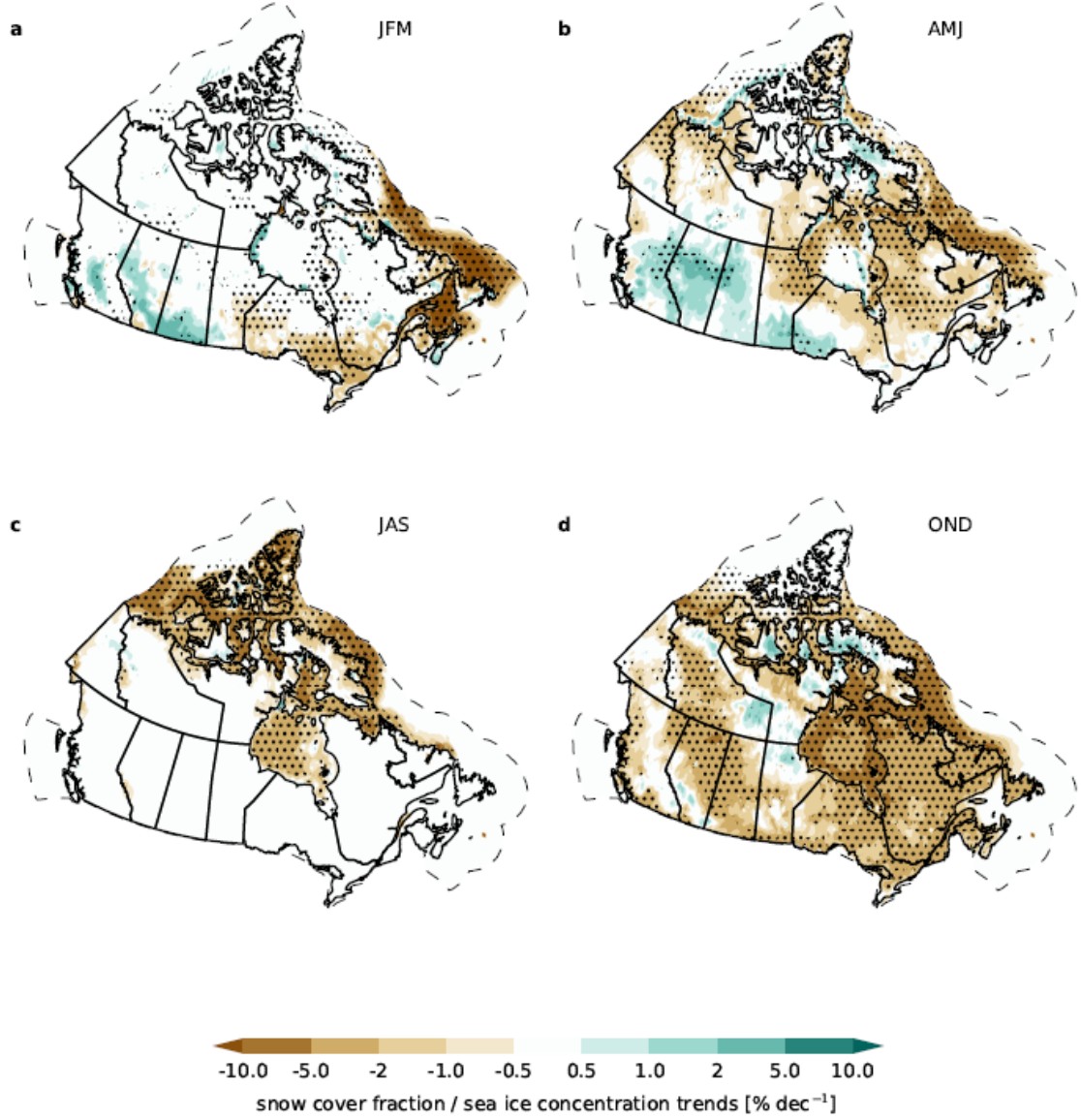

**Figure 1: Terrestrial snow cover fraction and sea ice concentration trends for 1981-2015. Datasets are described in Section 2.1. Stippling indicates pointwise significance at the 90th percentile. Dashed line denotes limit of Canadian marine territory.**

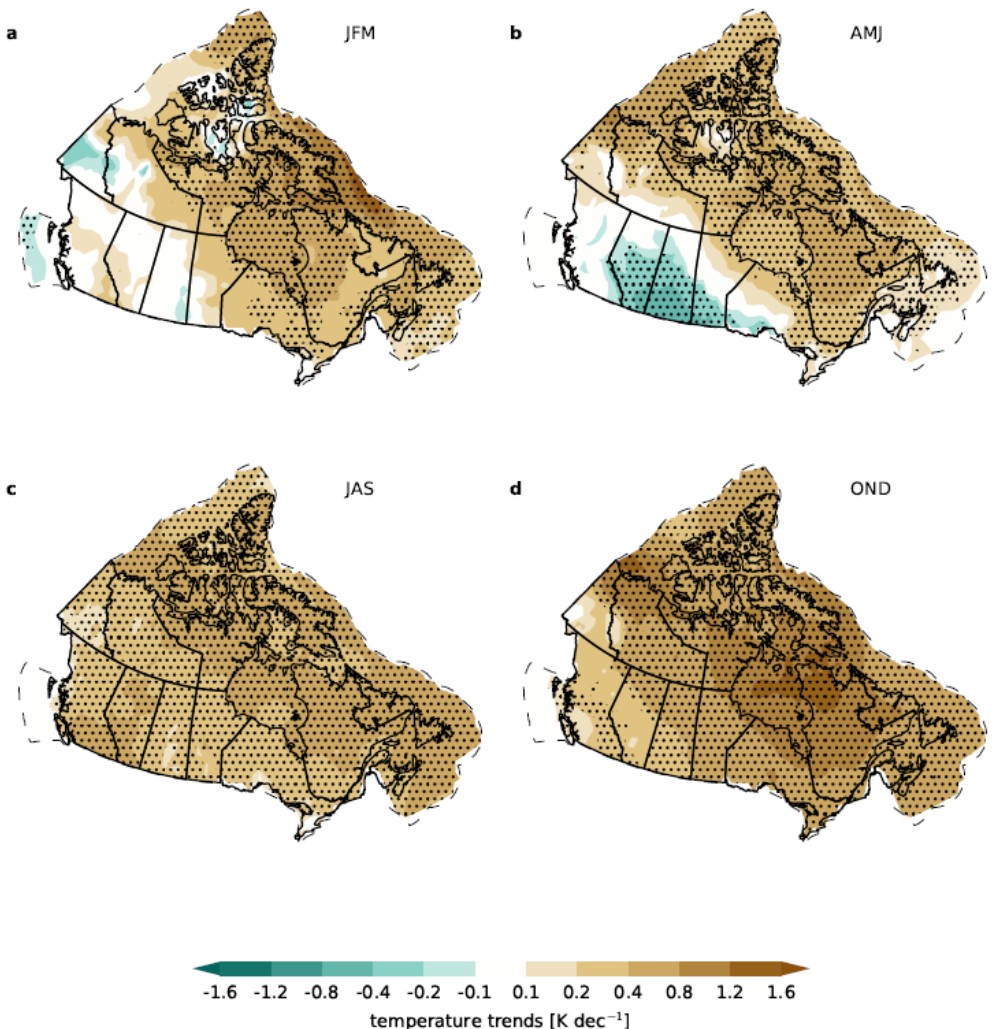

 **Figure 2: Trends in surface temperature, 1981-2015, from a blend of ERA-Interim, JRA-55, JRA-25, MERRA1, MERRA2, and CFSR reanalysis products. Stippling indicates pointwise significance at the 90th percentile.**

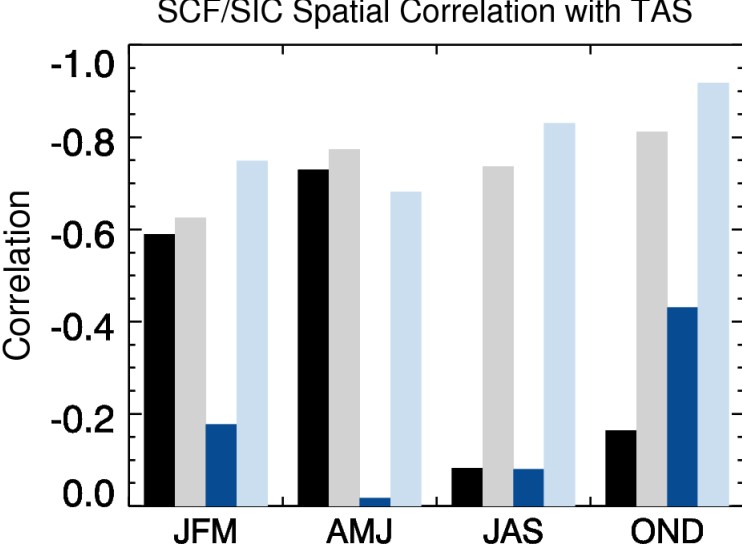

**Figure 3: Centered (dark) or uncentered (light) pattern correlation between seasonal TAS trends and seasonal SCF (black) or SIC (blue) trends.**

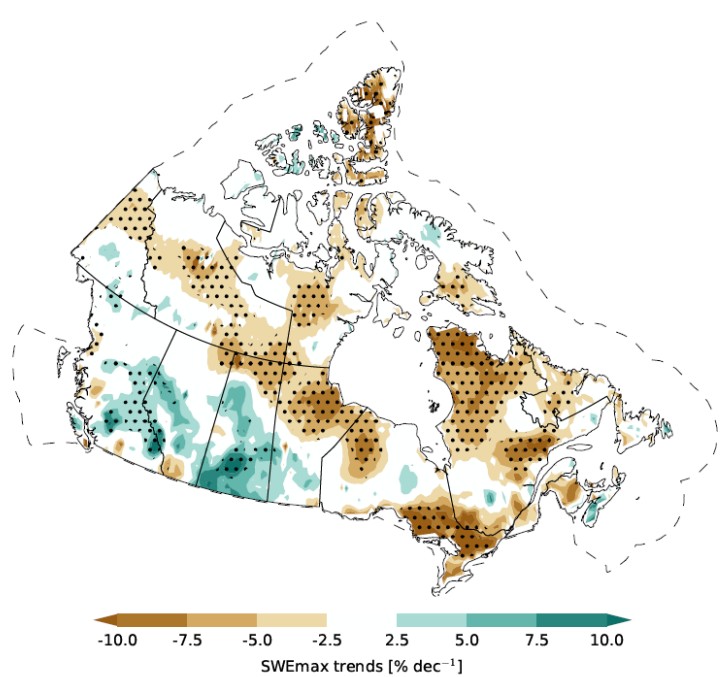

**Figure 4: SWEmax trends for 1981-2015. Stippling indicates pointwise significance at the 90th percentile.**

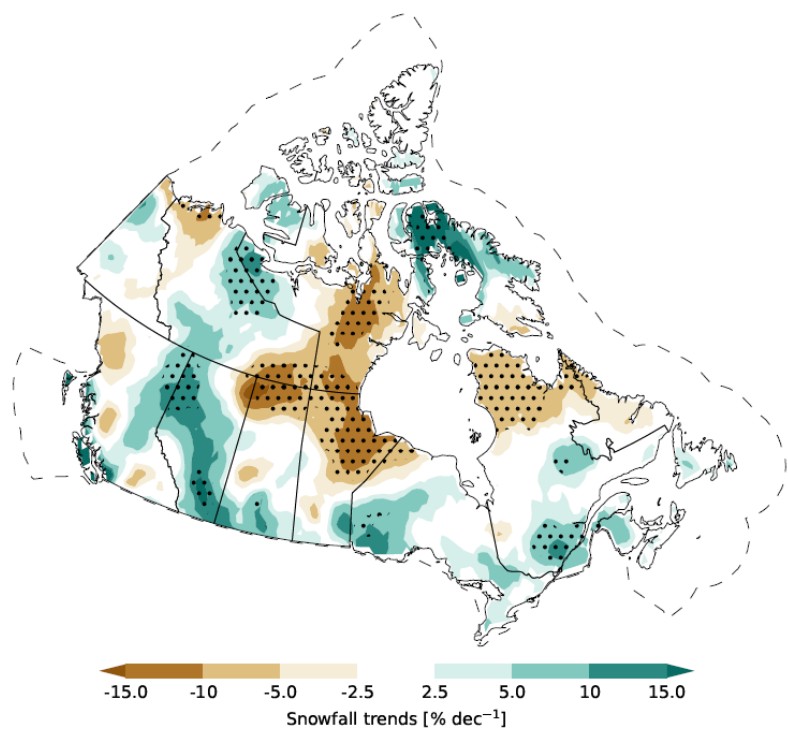

Figure 5: Snowfall trends estimated from CANGRD data. Stippling indicates pointwise significance at the 90th percentile.

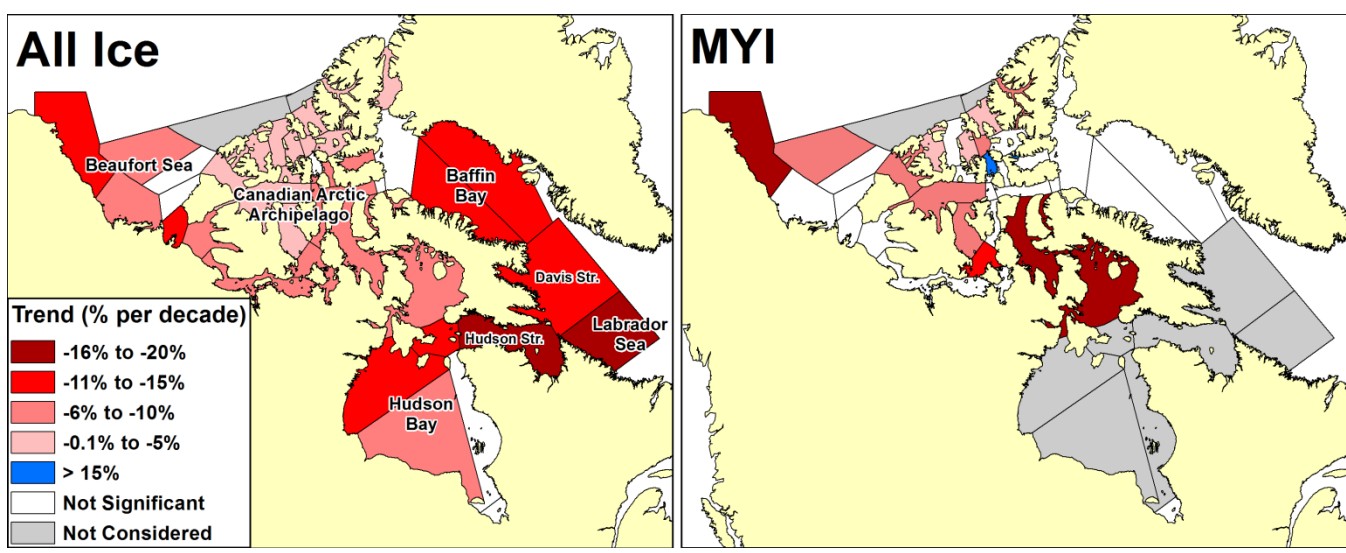

Figure 6: Trends in summer all ice (left) and multi-year ice (right) area from 1968 to 2016 from the CISDA. Only trends significant to the 95% confidence level are shown.

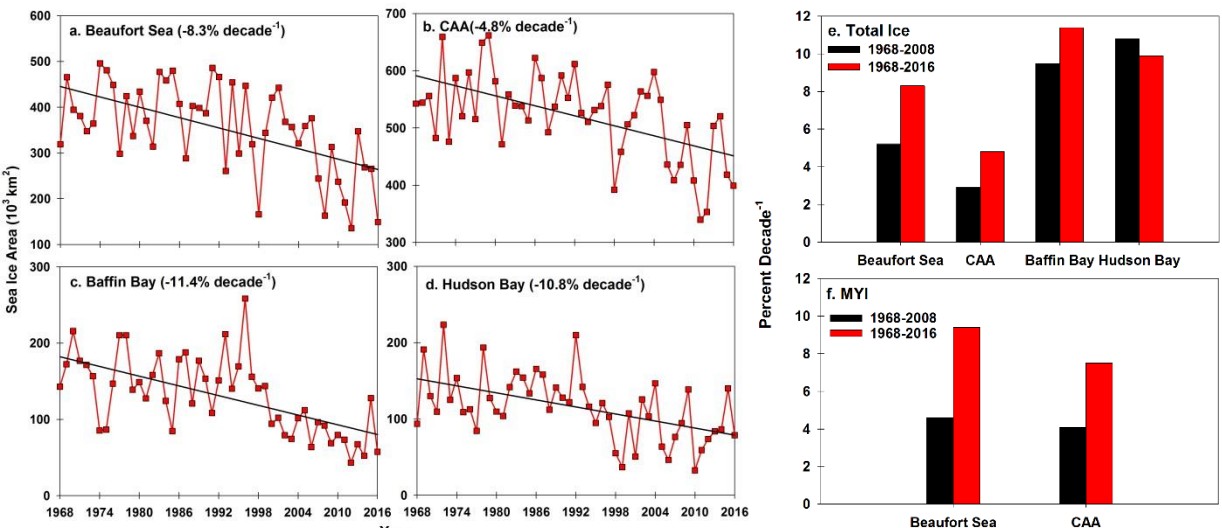

Figure 7: Time series of summer total sea ice area for the (a) Beaufort Sea, (b) Canadian Arctic Archipelago (CAA), (c) Baffin Bay and (d) Hudson Bay regions from 1968 to 2016. Comparison of trends between 1968-2008 and 1968-2016 for all ice (e) and multi-year ice (MYI) (f) for selected regions in the Canadian Arctic.

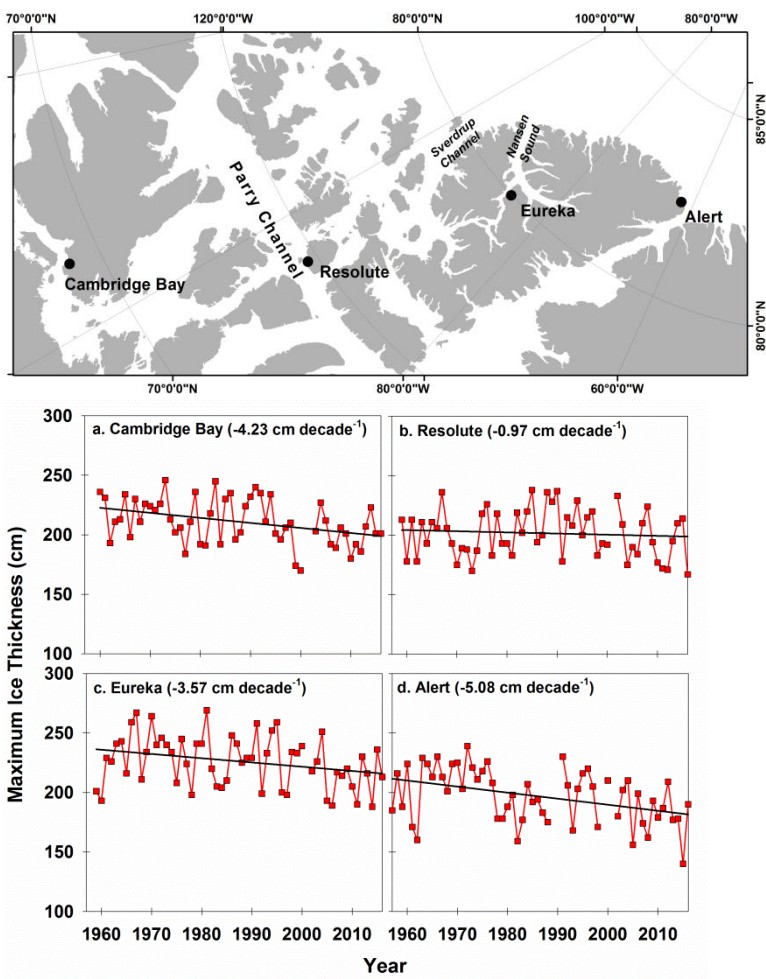

**Figure 8: Time series and trend of observed maximum ice thickness at (a) Cambridge Bay, (b) Resolute, (c) Eureka and (d) Alert locations in the Canadian Arctic.**

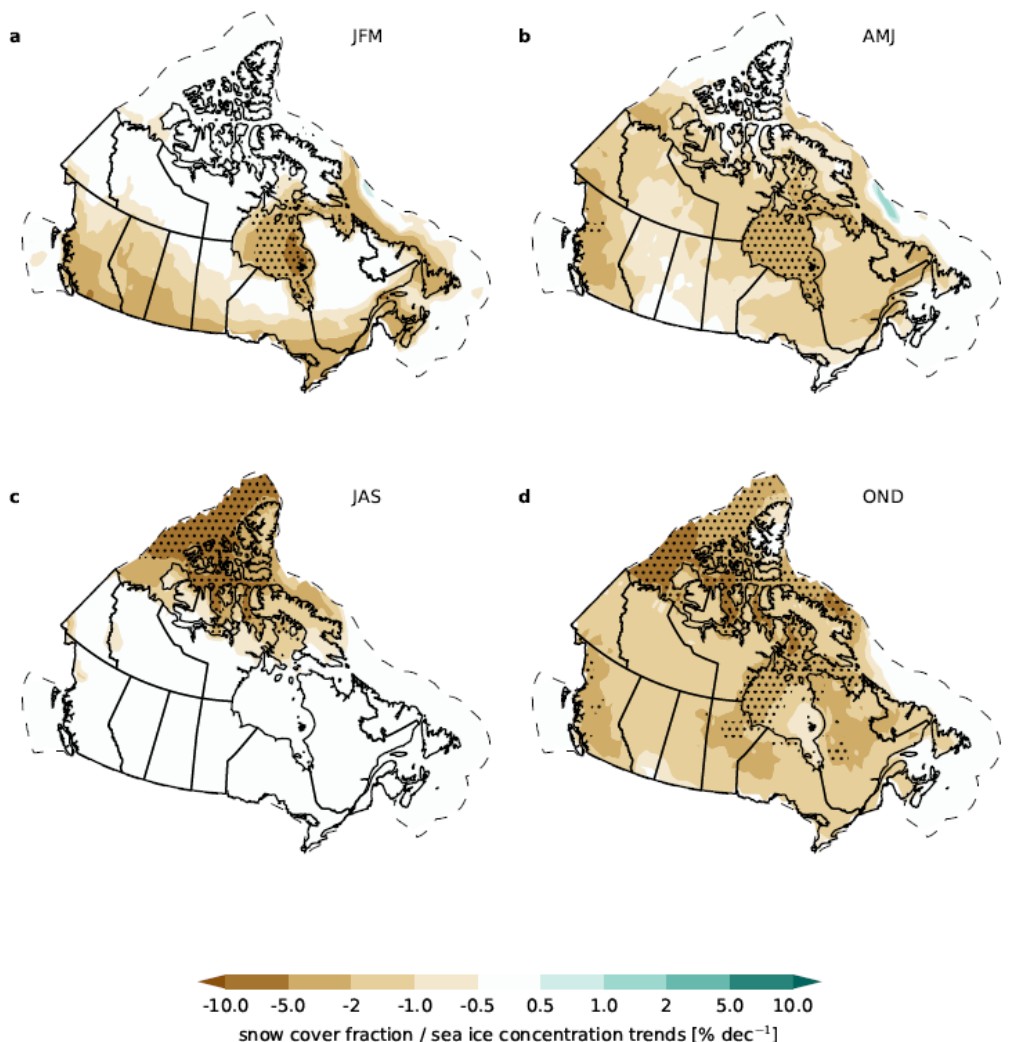

**Figure 9: Projected terrestrial snow cover fraction and sea ice concentration trends for 2020-2050. Model simulations are described in Section 2.2. Stippling indicates pointwise significance at the 90th percentile.**

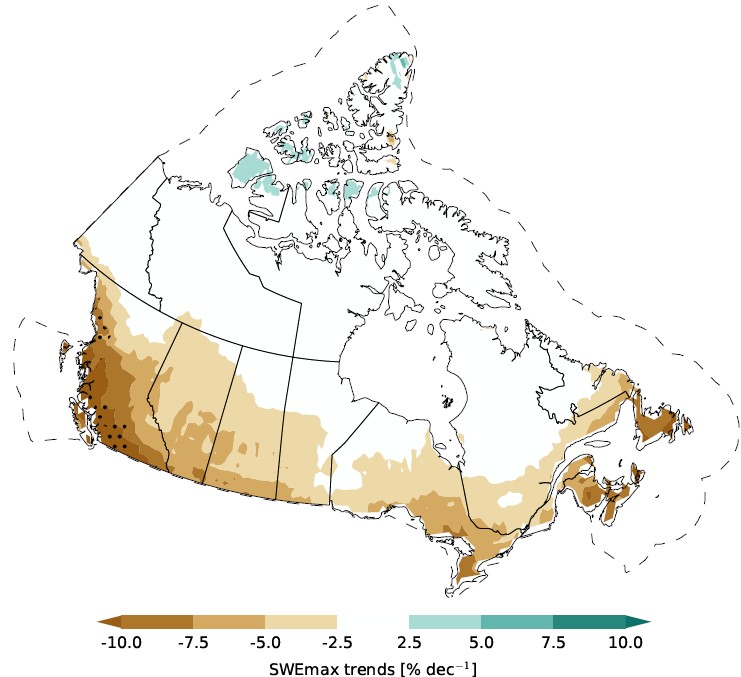

**Figure 10: 2020-2050 trends in SWEmax from the CMIP5 ensemble. Stippling indicates pointwise significance at the 90th percentile. We show percent change relative to the climatological (1981-2015) mean because there is large variability in SWEmax across the country (high SWE in western cordillera; low SWE in the Prairies).**

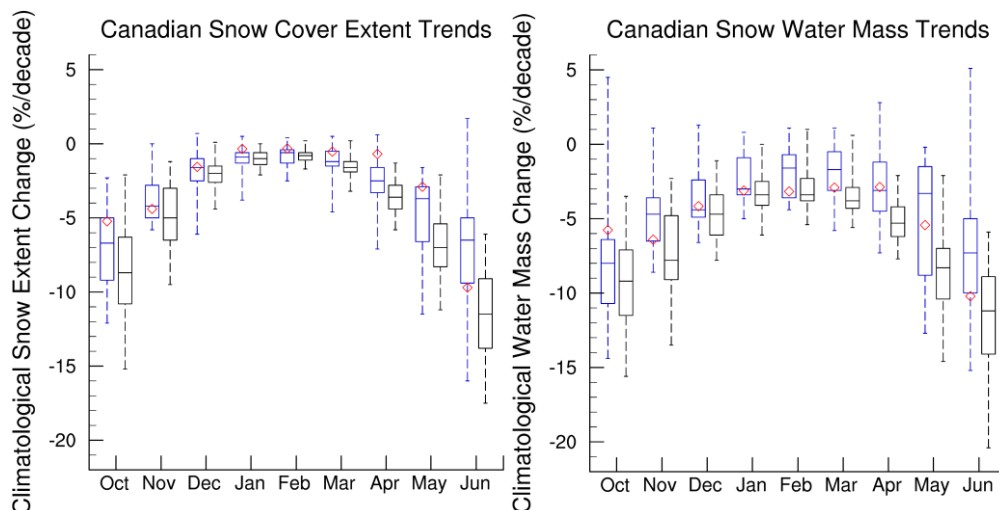

**Figure 11: 2020-2050 monthly trends in Canadian snow cover extent (left) and snow water mass (right) from the CMIP5 multimodel ensemble (blue) and CanESM large initial condition ensemble (black). Monthly mean observational trends (1981-2015) from the snow dataset used in Section 2 are shown in red. For each box the enclosed region shows the 25th-75th percentile range, the horizontal line shows the median, and the dashed whiskers illustrate the minimum and maximum.**

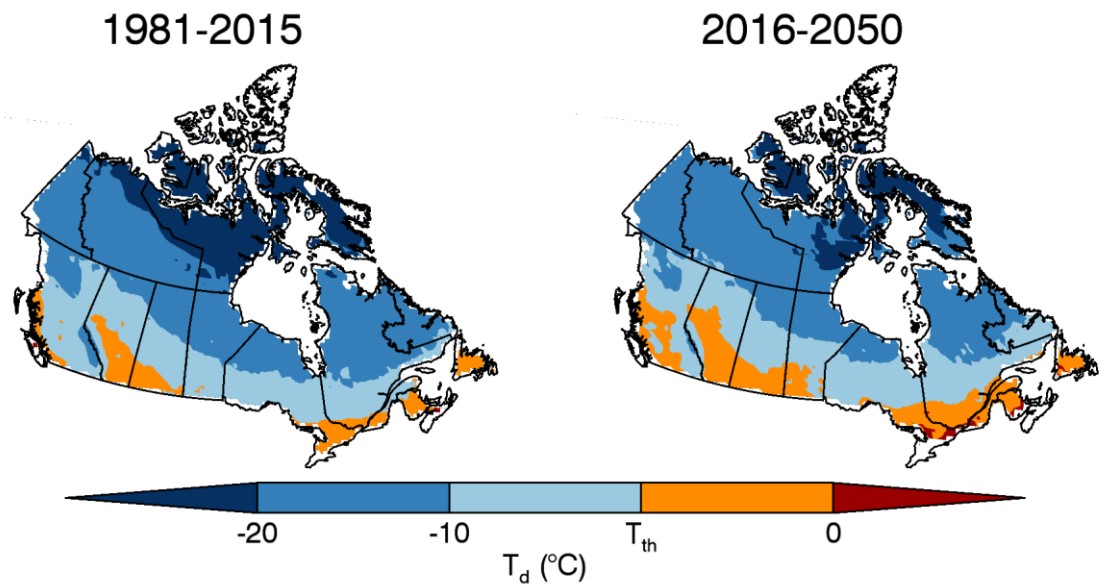

Figure 12: Temperature and precipitation controls on March snow water equivalent for historical period (left) and projections to 2050 (right). For the data presented here $T_{th}$ = -5.4°C (more generally $T_{th}$ = -5±1°C). Regions with $T_d < T_{th}$ (blue) have March SWE dominated by precipitation variability while regions with $T_d > T_{th}$ (orange) have March SWE dominated by temperature variability.

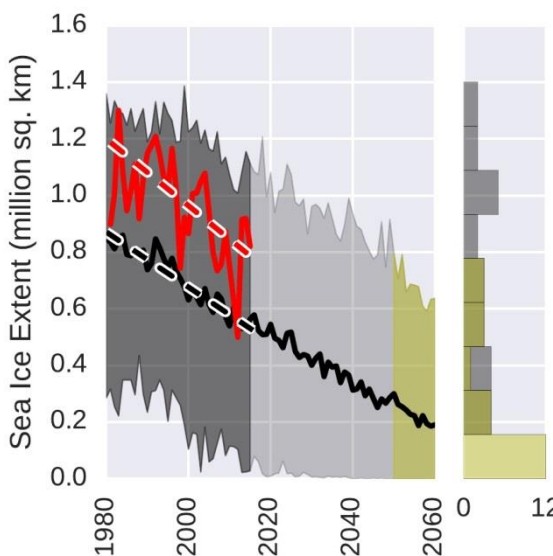

Figure 13: Observed (red line) September sea ice extent over Canadian marine regions (SIC > 0.15); CMIP-5 multi-model mean (black line) and spread (shading). Histogram on the right shows the model distribution for 1980-2015 in black and for 2050-2060 in yellow.

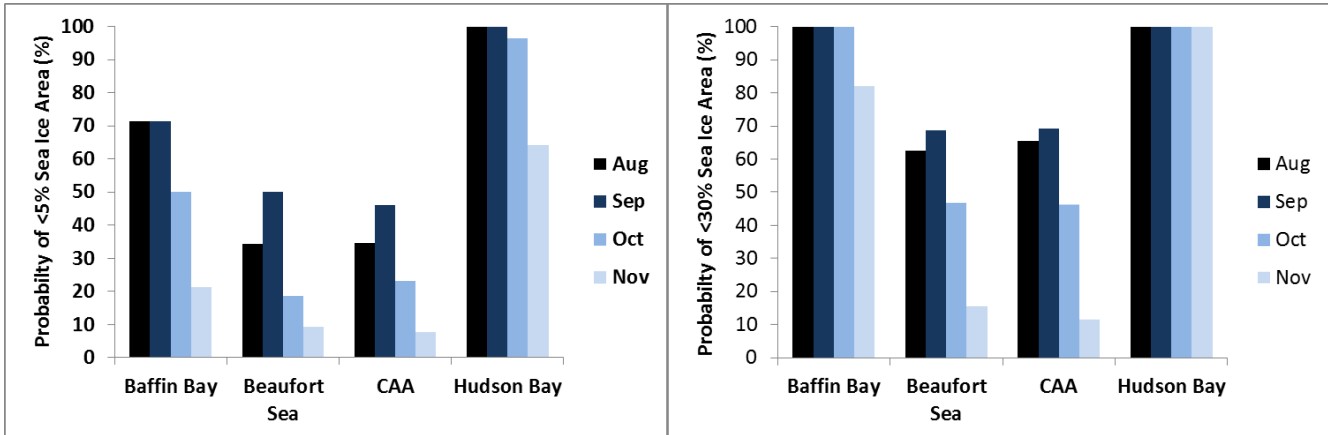

**Figure 14: Probability of sea ice free conditions by 2050 from the CMIP5 multi-model mean using a 5% (left) and 30% (right) sea ice area threshold.**