# Peer review of "Canadian Snow and Sea Ice: Historical Trends and Projections"

_The Cryosphere, 2017_

## Referee Comment (RC1) · C. Marty (Referee) · 22 Oct 2017

The current study analyses past and future snow and sea ice trends in Canada. For the past (1981-2015) the authors investigated snow trends from four different gridded data sets and sea ice trends from the CISD archive. For the future (2020-2050) they investigated the output of several dozens CMIP5 models. The results demonstrate clear changes for the past and the future despite some regional differences. Such analyses are important in a large northern region such as Canada because the corresponding changes in the surface energy budget may have a global impact. Moreover, such analyses are also an important basis to understand ecological and economical changes in such a sensitive region. The structure of the paper is clear, the text is concise and follows a obvious thread. The methods and results are nicely presented. Despite the

many acronyms the study is relatively easy to read. One point I'm missing is the fact that the precipitation impact is totally neglected in the analysis of the past snow trend. Otherwise, I have only some minor comments, which are listed below:

L29: You use at least four different expressions for annual maximum SWE throughout the paper. I don't like peak pre-melt SWE, since melt can also happen before maximum SWE. I suggest using always the same expression.

L 38: I would like to see some references, which demonstrate "the critical importance of snow/sea ice to Canada's natural environment, ecosystems, and economy."

L 99-104: Please write some sentences about the percentage of affected pixels and which regions are mainly impacted.

L 111-119: For consistency and understandability please introduce here somewhere the investigated sea ice variables, similarly as it has been done in the former chapter for snow.

L 116: As a non sea ice specialist, I would like to see short hint saying that sea ice concentration is a similar measure as snow cover fraction and is therefore also expressed in percentage.

L 136: Please provide some information about the spatial resolution of the used CMIP5 output.

L148-149: "...branch off from five different historical simulation... " I don't understand this sentence, please elaborate.

L 159: The acronym "SIE" is not used anymore in the following text, so please delete.

L 179-180: What is the reason you define the seasons different from the usual meteorological definition? Maybe, add a corresponding sentence in the methods chapter.

L 205: "...region with positive SCF trends is slightly more extensive." It would be helpful if you could at least write some sentences about the observed precipitation variability.

L 275: "modal thickness"?

L 283-284: The multi-model projected mean changes in surface temperature are positive in all seasons, hence only reductions in ensemble mean SCF and SIC are evident in Figure 8.

L 310: "... balances projected increases in snowfall." It's the first and only time you write about projected increases in snowfall. Could you please elaborate.

L 311: "snow mass", I guess you mean SWE, which would be a more familiar term and consistent with what you used already.

L 329: Is it not 1981-2015 for the present and 2020-2050 for the projection?

L 330: Why using a new variable Ts and not TAS?

L 349: ...shows the observed record of annual sea ice extent...

L 377: MYI losses

L 413-418: This paragraph seems kind of odd at the first sight, because alpine snow has only been mentioned once at the beginning. Please link this important content to the corresponding analysis.

L 427: FYI as not been introduced so far.

L 445: (Laliberté et al, 2016) "Resolving the dates...has important implications for climates studies" Please elaborate.

L622: Not in press anymore...

L 664-665: Please elaborate more clearly what's the difference between the region with single hatching and crossed hatching. The text in chapter 3.1 should also reflect theses differences.

L 685: I suggest to put the 10ˆ3 in the axis label for figure 6 a-d.

L 705: Replace snow water mass with snow water equivalent.

L 711: Please explain "Tsd" and mention that Tth is -5°C, since the scale of the legend bar is not linear. Also elaborate what Tth actually controls, i.e. that the bluish colored regions are mainly controlled by precipitation.

———————————————————

---

## Referee Comment (RC2) · Anonymous Referee #2 · 25 Oct 2017

This paper presents an assessment of historical (1981 onwards) and projected (until 2050) sea-ice and snow cover changes in Canada and the Canadian marine areas. It is well written and the methods applied are sound. The results are correctly and carefully interpreted and discussed in appropriate depth. The assessment does not provide any radically new insight into the field, but by looking at sea ice and continental snow cover at the same time, it offers an interesting comparison between the two domains. More-over, it offers some interesting points of view on some specific aspects such as North American snow cover data and sea-ice thickness in the Canadian Arctic Archipelago. Using in situ data to complement large-scale data in some specific cases and areas is a valuable idea.  The authors justify the exclusive focus on Canadian continental and marine areas by institutional reasons, and Canada (plus its marine areas, with or

without Hans Island) is clearly large enough for such a national assessment to make sense in an international publication, but at some places in the paper, it might have been interesting to put the results into a somewhat broader geographical context. The natural system does not care about national borders. I therefore think that the paper should be published after some minor changes suggested in the following.

Specific comments 1) Several of the figures were provided at rather low resolution, for example Figure 6. Please provide high-resolution figures in the end.

2) Line 35: If possible, you might consider providing the information about CANSISE in the acknowledgments rather than in the introduction; it is not of scientific interest per se.

3) Line 42: A more relevant reference for CMIP6 is Eyring et al., GMD 2016: https://www.geosci-model-dev.net/9/1937/2016/

4) Line 79: "future projections": climate projections are always for the future, I think. "climate projections" might be better

5) Line 159: "Laliberte" -> "Laliberté"

6) Line 187: "This difference may stem from...": It should be possible to evaluate how much of the discrepancy is due to not using the NOAA record. This might be an interesting information.

7) Line 205: "The reduced warming and cooling over northwestern Canada..." This sentence is a bit cryptic. Can you be a bit more explicit, e.g. explain what the Mydrik et al 2014 paper says?

8) Line 211-217: It is unclear to me why substracting a mean value should change the correlation – which means that I certainly haven't understood your explanation of centered and uncentered spatial correlations. Please consider giving some more detail on the method. What does an uncentered correlation mean physically?

9) Line 224: "…ice (…) is more difficult to melt than snow…" OK, one understands this of course, but the formulation sounds a bit strange.

10) Line 239 "the summer is ice season": delete "is"

11) Line 281: Relative impact of natural variability, scenario uncertainty and model uncertainty: Can you quantify this a bit better in this particular context?

12) Line 287: Typo ("reigon")

13) Line 302: OK, but SCF should also depend on precipitation rate, and probably more so than SIC does.

14) Line 324: Missing full stop at the end of the sentence. This result (that a large portion of the inter-model spread is associated with internal variability) is striking and might be emphasized/elaborated on a bit more.

15) Line 356: Full description in Laliberté et al., 2016: OK but can you give a very short description of the method here?

16) Line 361: "… will be sea ice free in September." Add, for clarity, "by 2050"; please indicate whether this concerns decadal averages, 30-year averages, extreme years,....

17) Line 388: "precipitation availability": sounds strange to me. Precipitation either occurs or not, it is not "available" for somebody to buy it.

---

## Author Comment (AC1) · 15 Jan 2018

All responses below refer to the revised version of the manuscript submitted as as supplementary pdf document (highlighted changes in red).

**L29: You use at least four different expressions for annual maximum SWE throughout the paper. I don't like peak pre-melt SWE, since melt can also happen before maximum SWE. I suggest using always the same expression.**

The expression annual maximum SWE, which we abbreviate as SWEmax, is now used throughout the paper.

**L 38: I would like to see some references, which demonstrate "the critical importance of snow/sea ice to Canada's natural environment, ecosystems, and econ-**

[Figure]

**omy."**

Text has been added at line 36:

**L 99-104: Please write some sentences about the percentage of affected pixels and which regions are mainly impacted.**

We have added additional information at line 106:

**L 111-119: For consistency and understandability please introduce here somewhere the investigated sea ice variables, similarly as it has been done in the former chapter for snow. L 116: As a non sea ice specialist, I would like to see short hint saying that sea ice concentration is a similar measure as snow cover fraction and is therefore also expressed in percentage.**

We have added additional text at lines 121 and 134.

**L 136: Please provide some information about the spatial resolution of the used CMIP5 output.**

Additional information has been provided at lines 155 and 171.

**L148-149: "...branch off from five different historical simulation..." I don't understand this sentence, please elaborate.**

The wording has been changed in lines 163-166. For the purpose of this paper it is only relevant that the ensemble is an initial condition ensemble. [Other ensembles of this sort have been constructed so that all realizations evolve under identical radiative forcings and using identical model parametrizations from small perturbations to a single climate state (for example, the NCAR initial condition ensemble). In the case of the CanESM2 initial condition ensemble, 5 interchangeable initial climate states were used. The initial states differed in that their exact climate tracks — their oceans in particular — had been allowed to diverge from one another over approximately 100 years. However, they are interchangeable because each was itself produced using identical radiative

forcings and identical model settings. As such they differ from one another only due to natural variability. Each of these 5 climate states was perturbed 10 times for a total of 50 realizations.]

**L 159: The acronym "SIE" is not used anymore in the following text, so please delete.**

Thank you. Removed.

**L 179-180: What is the reason you define the seasons different from the usual meteorological definition? Maybe, add a corresponding sentence in the methods chapter.**

Text added at line 195:

**L 205: "...region with positive SCF trends is slightly more extensive." It would be helpful if you could at least write some sentences about the observed precipitation variability.**

We now include an additional figure (figure 5) showing estimated annual snowfall trends and provide discussion starting at line 252:

**L 275: "modal thickness"?**

We have rephrased this sentence at line 303:

**L 283-284: The multi-model projected mean changes in surface temperature are positive in all seasons, hence only reductions in ensemble mean SCF and SIC are evident in Figure 8.**

Changed.

**L 310: "...balances projected increases in snowfall." It's the first and only time you write about projected increases in snowfall. Could you please elaborate.**

Because we don't have projections for snowfall we are unable to elaborate here, but

we have rephrased the sentence to make clear this is speculative (line 340).

**L 311: "snow mass", I guess you mean SWE, which would be a more familiar term and consistent with what you used already.**

We intended the term to distinguish the integrated quantity of snow water (its total volume or equivalently its mass) across a given area or region, which strictly speaking is not SWE. We have provided brief definitions for this term and snow cover extent which is defined similarly (line 343).

**L 329: Is it not 1981-2015 for the present and 2020-2050 for the projection?**

We have corrected the years for the historical period. The years included in the future period differ slightly from those used in the majority of the paper and are correct. We have removed the references in the title to specific time periods because along with minor differences in the future projections commented on here, some of the sea ice trends presented extend back to 1968.

**L 330: Why using a new variable Ts and not TAS?**

This section was unclear. Ts did not represent TAS as shown in Figure 2, it is a diagnostic temperature. We have revised the description at (lines 362-374) and relabeled the quantity as Td. Hopefully this will make the section clearer.

**L 349:...shows the observed record of annual sea ice extent...**

Changed.

**L 377: MYI losses**

Changed.

**L 413-418: This paragraph seems kind of odd at the first sight, because alpine snow has only been mentioned once at the beginning. Please link this important content to the corresponding analysis.**

We have tried to link this point more closely to the analysis contained in the paper (lines 457-461). The 2018 reference added is passed review so we are expecting to be able to update the reference before publication of this paper.

**L 427: FYI as not been introduced so far.**

Clarified.

**L 445: (Laliberté et al, 2016) "Resolving the dates...has important implications for climates studies" Please elaborate.**

This was rephrased. We meant only to say that knowledge of when the CAA will become ice free will facilitate adaptation (line 490).

**L622: Not in press anymore...**

Thanks!

**L 664-665: Please elaborate more clearly what's the difference between the region with single hatching and crossed hatching. The text in chapter 3.1 should also reflect these differences.**

We agree that the false discovery rate significance was unclear. We have decided that standard calculations of significance are sufficient. Captions and images for Figures 1,2,4, 9,10 have been altered to reflect this change.

**L 685: I suggest to put the $10^3$ in the axis label for figure 6 a-d.**

Good point. Changed.

**L 705: Replace snow water mass with snow water equivalent.**

Snow water mass has now been defined in the corresponding text.

**L 711: Please explain "$T_{sd}$" and mention that $T_{th}$ is $-5^oC$, since the scale of the legend bar is not linear. Also elaborate what $T_{th}$ actually controls, i.e. that the bluish colored regions are mainly controlled by precipitation.**

The text added in the manuscript should help clarify these points. Please note that $T_{sd}$ was a typo for the original variable label, $T_s$. We have renamed this variable $T_d$ in the revised manuscript for "diagnostic temperature." The caption now lists the value of $T_{th}$ and describes the different drivers of SWE variability in the different regions as requested.

Please also note the supplement to this comment:
https://www.the-cryosphere-discuss.net/tc-2017-198/tc-2017-198-AC1-supplement.pdf

[Figure]

**Supplement:**

[revised manuscript text omitted]

---

## Author Comment (AC2) · 15 Jan 2018

**Specific comments**

**1) Several of the figures were provided at rather low resolution, for example Figure 6. Please provide high-resolution figures in the end.**

Yes, these will be provided.

**2) Line 35: If possible, you might consider providing the information about CAN-SISE in the acknowledgments rather than in the introduction; it is not of scientific interest perse.**

Moved

**3) Line 42: A more relevant reference for CMIP6 is Eyring et al., GMD 2016: https://www.geosci-model-dev.net/9/1937/2016/**

Reference changed.

**4) Line 79: "future projections": climate projections are always for the future, I think. "climate projections" might be better**

Agreed (–> line 87).

**5) Line 159: "Laliberte" -> "Laliberté"**

Changed.

**6) Line 187: "This difference may stem from...": It should be possible to evaluate how much of the discrepancy is due to not using the NOAA record. This might be an interesting information.**

The plot attached illustrates both points during the spring (AMJ): 1) Arctic snow cover trends over Eurasia are stronger in both data sets than over North America; 2) the trend over the NH as a whole and over the NH Arctic region is about 50% stronger in the NOAA dataset than the multiple-source dataset used in our analysis. Both of these issues are discussed in detail in the Mudryk et al 2017 citation listed on line 209.

Caption: AMJ SCF trends in multi-source dataset (left) and NOAA dataset for 1981-2015 period. Pairs of numbers on the far right indicate the average trend over the entire NH (black), NH Arctic (blue) and Canada (red) for the multisource dataset (top) and the NOAA dataset (bottom).

**7) Line 205: "The reduced warming and cooling over northwestern Canada..." This sentence is a bit cryptic. Can you be a bit more explicit, e.g. explain what the Mudryk et al 2014 paper says?**

We have added text at lines 224-229.

**8) Line 211-217: It is unclear to me why subtracting a mean value should change the correlation – which means that I certainly haven't understood your explanation of centered and uncentered spatial correlations. Please consider giving some more detail on the method. What does an uncentered correlation mean physically?**

The Peason's correlation coefficient that is typically calculated *is* a centered correlation (the mean field value is subtracted before the fields are multiplied). For an uncentered correlation, the mean value is not removed. This metric measures the average behavior of the fields (whether they are both positive on average, both negative on average, of opposite sign on average, or unrelated on average). Because spatial patterns of SIC and SCF trends may fluctuate in response to multiple drivers other than TAS, the uncentered correlation provides confirmation that overall, both fields are decreasing in response to increasing surface temperature (as expected), however, there may be differences resulting from other drivers.

**9) Line 224: "...ice (...) is more difficult to melt than snow..." OK, one understands this of course, but the formulation sounds a bit strange.**

Wording clarified on line 245.

**10) Line 239 "the summer is ice season": delete "is"**

Thanks.

**11) Line 281: Relative impact of natural variability, scenario uncertainty and model uncertainty: Can you quantify this a bit better in this particular context?**

We have changed the citation to Hawkins and Sutton, 2009 and 2011 which allows us to cite their estimates of the fraction of uncertainty due to forcing scenario at a lead time appropriate for mid-century projections (lines 310-313).

**12) Line 287: Typo ("reigon")**

I couldn't find this typo. Perhaps it got changed at some point.

**13) Line 302: OK, but SCF should also depend on precipitation rate, and probably more so than SIC does.**

As part of the response to reviewer 1, we have included additional analysis on snowfall trends (new Figure 5) and new text at lines: 252-263.

**14) Line 324: Missing full stop at the end of the sentence. This result (that a large portion of the inter-model spread is associated with internal variability) is striking and might be emphasized/elaborated on a bit more.**

We have added an additional reference at line 356.

**15) Line 356: Full description in Laliberté et al., 2016: OK but can you give a very short description of the method here?**

Included (lines 394-399).

**16) Line 361: "...will be sea ice free in September." Add, for clarity, "by 2050"; please indicate whether this concerns decadal averages, 30-year averages, extreme years,....**

Added "by 2050". The additional description should address the latter portion of the comment.

**17) Line 388: "precipitation availability": sounds strange to me. Precipitation either occurs or not, it is not "available" for somebody to buy it.**

This was a typo. We have rephrased the sentence (line 430).

Please also note the supplement to this comment:
https://www.the-cryosphere-discuss.net/tc-2017-198/tc-2017-198-AC2-supplement.pdf

[Figure]

[Figure]

**Fig. 1.** AMJ SCF trends

**Supplement:**

[revised manuscript text omitted]